# OpenReview forum: "Obscure but Effective: Classical Chinese Jailbreak Prompt Optimization via Bio-Inspired Search"
_ICLR.cc/2026/Conference — ICLR 2026 Poster_

### Official Review · Reviewer_hZfy · 2025-10-25

**Soundness:** 3
**Presentation:** 2
**Contribution:** 2
**Rating:** 4
**Confidence:** 4

**Summary:**

The work proposes CC-BOS, a black-box jailbreak framework for Large Language Models (LLMs) using Classical Chinese. It leverages Classical Chinese’s linguistic traits (conciseness, obscurity, etc.) to bypass safety constraints, structures prompt generation into an eight-dimensional strategy space, and adopts a fruit fly-inspired bio-optimization algorithm for efficient space exploration. Experiments demonstrate the performace of the proposed algorithm.

**Strengths:**

1. The proposed jailbreak method achieves the best attack performance on AdvBench Benchmark compared with existing methods.
2. The proposed jailbreak method can achieve success with a relatively small number of queries, which takes into account the efficiency of the algorithm.

**Weaknesses:**

1. The innovation and contributions of this paper are relatively limited. Jailbreak attack methods based on genetic algorithms have already been proposed (e.g. AutoDAN). Although this paper puts forward some different strategy spaces, their contributions are quite limited.
2. The adversarial prompt examples provided in Figures 3 and 4 in appendix contain prior information about malicious queries, such as the requirement of potassium nitrate for making bombs. This indicates that additional prior information about queries is incorporated into the attack process, making the comparison less fair.
3. Since this paper claims that classical Chinese can easily bypass the model's safety safeguards, I am very curious about the possible reasons for this phenomenon, as the safety and alignment of large models do not seem to have a direct connection with classical Chinese. I think this will make the method more interpretable.

**Questions:**

See weaknesses

---

> ### Author Response · Authors · 2025-11-20
> **Rebuttal by Authors[1/2]**
>
> Thank you for your valuable review and suggestions. Below we respond to the comments in Weaknesses (W) and Questions (Q).
>
> ***W1: The innovation and contributions of this paper are relatively limited. Jailbreak attack methods based on genetic algorithms have already been proposed (e.g. AutoDAN). Although this paper puts forward some different strategy spaces, their contributions are quite limited.***
>
> We appreciate your comment but respectfully clarify that CC-BOS is fundamentally different from Genetic Algorithm-based methods (like AutoDAN) in terms of both optimization granularity and algorithmic mechanism. It is not a simple application of GA.
>
> **1. Optimization Granularity (Strategy vs. Token):**
> Unlike AutoDAN, which optimizes discrete text token sequences directly via methods like Genetic Algorithms, CC-BOS optimizes an abstract 8-dimensional strategy vector. This decoupled approach searches for high-level semantic attack strategies rather than manipulating discrete token suffixes.
>
> **2. Algorithm Efficiency (FOA vs. GA):**
> We specifically utilize the bio-inspired optimization algorithm (BIO), which draws inspiration from the Fruit Fly Optimization Algorithm, not a generic GA. The inherent smell and vision search mechanisms of BIO provide a highly suitable and efficient exploration strategy for our discrete space. As shown in the table below, our method achieves 100% ASR with only 1.28 queries, significantly outperforming the standard GA (94% ASR, 4.04 queries).
>
> | Optimizer | ASR (%) | Avg. Q |
> |-----------|---------|---------|
> | **FOA (Ours)** | **100%** | **1.28** |
> | Genetic Algorithm (GA) | 94.00% | 4.04 |
> | Random Search | 90.00% | 6.10 |
>
> **Innovation and Contributions:**
> We respectfully disagree with the assessment that our novelty is limited. Our work, CC-BOS, is the first to leverage the novel linguistic environment of Classical Chinese (CC), define a systematic 8D strategy space based on its characteristics, and introduce an efficient optimization framework to search for the optimal strategy combination. Specifically:
>
> 1. **Vulnerability in Classical Chinese Context:**
>    We are the first to introduce the Classical Chinese context into adversarial prompt generation, systematically revealing a major, previously unstudied, vulnerability in LLM security.
>
> 2. **Systematic Strategy Space based on Classical Chinese Features:**
>    We propose a black-box jailbreak framework that formalizes prompt generation within an eight-dimensional strategy space. This space is a fusion of existing jailbreak strategies and the contextual properties of Classical Chinese, covering dimensions like role, behavior, mechanism, and metaphor.
>
> 3. **Bio-Inspired Optimization (BIO) Algorithm:**
>    To efficiently explore this multi-dimensional space, we leverage a bio-inspired optimization algorithm inspired by fruit fly foraging behavior. The algorithm integrates smell search, visual search, and cauchy mutation operators to facilitate efficient and automated iterative refinement of the prompt-generation strategy in black-box settings.
>
> 4. **State-of-the-Art (SOTA) Performance:**
>    Extensive experiments demonstrate that the proposed CC-BOS consistently outperforms state-of-the-art jailbreak attack methods. CC-BOS achieves a nearly 100% attack success rate (ASR) across all six mainstream LLMs.
>
> Our work provides a systematic, efficient framework for generating adversarial prompts based on Classical Chinese context, significantly advancing LLM safety research and robustness testing. We hope that reviewers can reassess our contributions from the perspective of advancing LLM safety research and robustness testing.

---

> ### Author Response · Authors · 2025-11-20
> **Rebuttal by Authors [2/2]**
>
> ***W2: The adversarial prompt examples provided in Figures 3 and 4 in appendix contain prior information about malicious queries, such as the requirement of potassium nitrate for making bombs. This indicates that additional prior information about queries is incorporated into the attack process, making the comparison less fair.***
>
> We appreciate your close scrutiny. We address this via two key distinctions:
>
> **1. Distinction between Domain Concepts and Actionable Details:**
> As shown in Figure 3, our prompts only mention domain concepts (like "Saltpeter") as necessary elements to construct a specific scenario (e.g., "ancient alchemy"). Crucially, the prompt does not contain specific ratios or manufacturing procedures.
>
> - **Our Query:** Mentions raw materials in an archaic context (e.g., "If using saltpeter as White Tiger... how should the numbers of the Nine Palaces match?"). It solicits the methodology rather than supplying it.
> - **Target Model Response:** The model replies with detailed chemical ratios and step-by-step fabrication processes.
>
> This confirms that the harmful information (the "how-to" knowledge) is entirely generated by the Target Model, not provided in the input. This is the hallmark of a successful jailbreak: using innocuous terminology to induce the release of harmful actionable instructions.
>
> **2. Details are Autonomously Generated via Strategy (Not Human Injection):**
> These specific terms are not manually injected prior knowledge. They are the autonomous output of the Attack LLM, guided by our 8D strategy. The Attack LLM automatically utilizes its pre-trained knowledge to associate the malicious intent (e.g., "Bomb") with relevant domain terms (e.g., "Saltpeter"), thereby generating prompts that exhibit high contextual congruence and elicitation power.
>
> Therefore, Figures 3 and 4 demonstrate precisely how CC-BOS constructs a high-credibility specialized context to successfully induce the model into revealing dangerous knowledge, rather than directly leaking the solution in the input.
>
> ---
>
> ***W3: Since this paper claims that classical Chinese can easily bypass the model's safety safeguards, I am very curious about the possible reasons for this phenomenon, as the safety and alignment of large models do not seem to have a direct connection with classical Chinese. I think this will make the method more interpretable.***
>
> We entirely agree that elucidating the underlying mechanism is crucial for the interpretability of our method. We attribute the effectiveness of Classical Chinese (CC) for bypassing LLM safety guardrails to three primary mechanisms:
>
> **1. The Capability-Alignment Mismatch:**
> During pre-training, LLMs ingest vast corpora of classical texts (e.g., historical records, military treatises, medical canons), endowing them with strong comprehension capabilities in Classical Chinese. Conversely, existing safety alignment predominantly focuses on Modern Chinese or English. This creates a defense blind spot: while the model fully comprehends the harmful intent within the archaic text, the safety mechanisms, which are trained on modern data, fail to recognize these archaic formulations as violations.
>
> **2. Linguistic Camouflage:**
> Classical Chinese inherently possesses high polysemy (multiple meanings per word) and relies on high-level metaphors (as noted in the paper, "expressing intent through objects," e.g., describing chemical reactions as "harmonizing Yin and Yang"). This dual mechanism creates a deep semantic blind spot: the polysemy introduces word-level ambiguity that complicates intent recognition, while the metaphor elevates specific, low-level harmful acts into benign-sounding, abstract concepts. This abstraction evades safety mechanisms trained to detect concrete, modern expressions of harm.
>
> **3. Contextual Trust Bias:**
> Classical Chinese is statistically associated with high-authority, benign domains such as history, literature, and philosophy. By constructing contexts like "Scholars discussing the Dao," the model perceives the prompt as a "knowledge retrieval" or "literary creation" task rather than a malicious instruction. This Benign Context Masquerade significantly lowers the model's defensive vigilance.

---

> ### Author Response · Authors · 2025-11-27
> **Looking forward to further feedback**
>
> Dear Reviewer hZfy,
>
> Sorry for bothering you, but the discussion period is coming to an end in a week. Could you please let us know if our responses have alleviated your concerns? If there are any further comments, we will do our best to respond.
>
> Best,
>
> The Authors

---

### Official Review · Reviewer_Qb4q · 2025-11-01

**Soundness:** 2
**Presentation:** 2
**Contribution:** 2
**Rating:** 2
**Confidence:** 5

**Summary:**

The authors propose a bio-inspired optimization algorithm, inspired by fruit fly foraging behavior, which enables automated and effective generation of adversarial prompts by balancing global exploration and local
exploitation.

They  design a two-stage translation module to ensure a more objective
and robust evaluation of model responses. By integrating these components, they develop a highperformance and stable jailbreaking method.


They conduct extensive experiments across multiple
LLMs to validate the effectiveness of our proposed method. The results demonstrate that CC-BOS
consistently outperforms existing jailbreak methods in success rate.

**Strengths:**

They propose classical Chinese into the study of adversarial prompt generation and jailbreaks for
the first time, thereby establishing a new perspective and extending the scope of LLM security.

They propose a black-box jailbreak framework that formalizes prompt generation within an eightdimensional strategy space and leverages the bio-inspired optimization algorithm to achieve
systematic and automated jailbreak prompt generation.

They construct a two-stage translation module to progressively mitigate the metaphorical and semantically compressed characteristics of classical Chinese, ensuring consistency and reliability
in the model response evaluation process.

**Weaknesses:**

1. The paper investigates the role of classical Chinese in jailbreak attacks. Can you generalize your paper technique into many other different languages?

2. After reading the paper, I cannot understand how you model the  jailbreak attacks with the feature of classical Chinese?

3. Why the BIO-INSPIRED OPTIMIZATION ALGORITHM is effective for classical Chinese for jailbreak attacks is very unclear to me.

**Questions:**

see the above comments.

---

> ### Author Response · Authors · 2025-11-20
> **Rebuttal by Authors [1/2]**
>
> Thank you for your valuable review and suggestions. Below we respond to the comments in Weaknesses (W) and Questions (Q).
>
> ***W1: The paper investigates the role of classical Chinese in jailbreak attacks. Can you generalize your paper technique into many other different languages?***
>
> Thank you for this question. We emphatically state that the effectiveness of our method is by no means limited to Classical Chinese; it can extend to other "classic languages" that possess rich historical corpora but lack sufficient safety alignment.
>
> To validate this, we extended the CC-BOS framework to Latin and Sanskrit and evaluated them on three state-of-the-art models (Gemini-2.5-flash, GPT-4o, and Deepseek-Reasoner). As shown in Table below, our method maintains an ASR of over 94% across all settings. Notably, attacks using Latin achieved a 100% success rate on both GPT-4o and Deepseek-Reasoner.
>
> These results confirm a critical structural vulnerability we term **"High Capability - Low Alignment"**. The classical languages, specifically Classical Chinese, Latin, and Sanskrit, constitute a portion of LLM pre-training corpora. Consequently, models exhibit robust understanding and generation capabilities in these languages (**High Capability**). However, existing safety alignment strategies are predominantly tailored to modern languages. This discrepancy leads to a critical misalignment. While the model fully comprehends malicious instructions formulated in these classical languages, its safety guardrails are rendered ineffective due to a lack of exposure to these specific linguistic distributions during the alignment phase.
>
> In summary, it exposes a universal vulnerability in how LLMs handle these classic languages. We have integrated these experiments into Appendix [I] of the revision.
>
> | Target Model | Language | ASR |
> |--------------|----------|-----|
> | Gemini-2.5-flash | Classical Chinese | 100% |
> |  | Latin | 96% |
> |  | Sanskrit | 98% |
> | GPT-4o | Classical Chinese | 100% |
> |  | Latin | 100% |
> |  | Sanskrit | 94% |
> | Deepseek-Reasoner | Classical Chinese | 100% |
> |  | Latin | 100% |
> |  | Sanskrit | 98% |

---

> ### Author Response · Authors · 2025-11-20
> **Rebuttal by Authors [2/2]**
>
> ***W2: After reading the paper, I cannot understand how you model the jailbreak attacks with the feature of classical Chinese?***
>
> Thank you for this question. Our core contribution is formalizing the unique linguistic properties of Classical Chinese (CC) into a computable, optimizable 8-dimensional strategy space. As defined in Eq. (5) of our paper, we model the attack vector as a Cartesian product of these dimensions.
>
> Specifically, we map the linguistic characteristics of Classical Chinese to our model dimensions as follows (detailed in Appendix C.2):
>
> 1. **Semantic Obscurity & Metaphor -- Dimensions: [Metaphor Mapping], [Mechanism], & [Knowledge Relation]**
>    - *Linguistic Feature:* Classical Chinese frequently employs "expressing intent through objects" to circumvent explicit articulation.
>    - *Modeling:* We utilize Metaphor Mapping to replace sensitive terms (e.g., "cyberattack") with classical metaphors (e.g., "The Art of War's principles of reality and illusion"); design obscure induction mechanisms via Mechanism (e.g., "prophecy and apocrypha"); and use Knowledge Relation to reconstruct modern technical concepts into ancient knowledge systems (e.g., mapping "Eight Trigrams positions" to "memory addresses"). These three dimensions collectively establish a deep semantic labyrinth, directly exploiting obscurity to bypass keyword-based defenses.
>
> 2. **Cultural Authority & High Context -- Dimensions: [Role Identity] & [Contextual Setting]**
>    - *Linguistic Feature:* Classical Chinese texts typically originate from historical canons, possessing an inherent sense of authority.
>    - *Modeling:* We construct a high-credibility historical context through Role Identity (e.g., "Grand Confucian Scholar", "Director of the Grand Divination Office") and Contextual Setting (e.g., "Court Policy Debate"). This exploits the model's "trust bias" towards classical literature, compelling it to switch from a "safety auditing mode" to a "historical knowledge retrieval mode," thereby lowering defensive vigilance.
>
> 3. **Syntactic Structure -- Dimensions: [Expression Style], [Trigger Pattern] & [Behavioral Guidance]**
>    - *Linguistic Feature:* Classical Chinese utilizes specific structures like "Parallel Prose" and concise grammar.
>    - *Modeling:* We exploit this by using Expression Style to mandate the model's generation of these atypical syntaxes; utilizing Trigger Pattern to control the timing of activation (e.g., "foreshadowing and activation"); and integrating Behavioral Guidance (e.g.,"cognitive confusion") to construct complex logical traps.
>
> The necessity of this modeling is empirically proven in our Ablation Study (Table 6).
> - Using translated Classical Chinese text alone (CC Base) yields only **18% ASR**.
> - Applying our 8-dimensional modeling (CC + Strategy) jumps the ASR to **60%**.
>
> This demonstrates that it is not just the "language" of Classical Chinese, but our structured modeling of its adversarial features that unlocks the jailbreak potential.
>
> We hope this clarification elucidates how we model the jailbreak attacks with the feature of classical Chinese.
>
> ---
>
> ***W3: Why the BIO-INSPIRED OPTIMIZATION ALGORITHM is effective for classical Chinese for jailbreak attacks is very unclear to me.***
>
> The selection of the BIO algorithm was a deliberate design choice, drawing inspiration from the Fruit Fly Optimization Algorithm. This approach is specifically utilized to efficiently explore our constructed 8-dimensional optimization space and identify the optimal combination.
>
> **1. Adaptability to the Search Space (Discrete & High-Dimensional):**
> The 8-dimensional strategy space constructed in our work essentially presents a large-scale discrete combinatorial optimization problem. BIO is a gradient-free, black-box heuristic algorithm. It does not require knowledge of the model's internal parameters and can efficiently search for the optimal combination within the discrete 8-dimensional strategy space. As shown in the table below, our method achieves 100% ASR with only 1.28 queries, significantly outperforming the GA and Random Search.
>
> | Optimizer | ASR | Avg. Q |
> |-----------|------|---------|
> | **FOA (Ours)** | **100%** | **1.28** |
> | Genetic Algorithm (GA) | 94.00% | 4.04 |
> | Random Search | 90.00% | 6.10 |
>
> **2. Robust Empirical Support:**
> Our ablation studies directly substantiate the necessity of this optimization (refer to Table 6): While using Classical Chinese alone (Base) yielded an ASR of only 18%, and employing a fixed strategy (CC + Strategy) increased the ASR to 60%, the introduction of the BIO optimization algorithm resulted in the ASR surging to a perfect 100%.
>
> In conclusion, the efficacy of BIO stems from its efficient search mechanism, which enables the rapid identification of specific combinatorial patterns within the 8D space that successfully trigger model vulnerabilities.

---

> ### Author Response · Authors · 2025-11-27
> **Looking forward to further feedback**
>
> Dear Reviewer Qb4q,
>
> Sorry for bothering you, but the discussion period is coming to an end in a week. Could you please let us know if our responses have alleviated your concerns? If there are any further comments, we will do our best to respond.
>
> Best,
>
> The Authors

---

### Official Review · Reviewer_kUsg · 2025-11-01

**Soundness:** 3
**Presentation:** 3
**Contribution:** 3
**Rating:** 6
**Confidence:** 4

**Summary:**

This paper proposes a novel jailbreak attack method. It leverages the linguistic characteristics of classical Chinese and introduces a framework, CC-BOS, for automatic generation of jailbreak prompts based on a multi-dimensional Fruit Fly Optimization algorithm.

**Strengths:**

1. The paper identifies a previously underexplored vulnerability of LLMs to jailbreaks that arises from classical Chinese, and proposes an automated method to generate jailbreak prompts.
2. The attack is evaluated against six baselines; experimental results indicate the proposed method is effective.

**Weaknesses:**

The experimental evaluation of the proposed attack against defenses is insufficient: only a single defense method is assessed. The paper should evaluate additional defense methods (e.g., composite and dynamic defenses) to support its claims.

**Questions:**

1. The paper claims that "the security vulnerabilities of classical Chinese stem from the unique challenges posed by its linguistic characteristics to alignment mechanisms". This viewpoint is confusing. The authors should provide more experiments and a deeper analysis to substantiate this claim. Yet the proposed method depends on translating classical Chinese as part of the jailbreak process. If alignment on classical Chinese is challenging, why does the LLM still exhibit strong translation capability?
2. The attack cost reported in Table 3 is unclear. The table reports an average success at ~1.5 queries, but according to the described framework, generating a single jailbreak prompt requires at least 4 interactions with the LLM in a single round. Please recheck how this metric is computed, clarify the calculation procedure, and update the results.

---

> ### Author Response · Authors · 2025-11-20
> **Rebuttal by Authors**
>
> Thank you for your valuable review and suggestions. Below we respond to the comments in Weaknesses (W) and Questions (Q).
>
> ***W1: The experimental evaluation of defense measures is insufficient: only one defense method was evaluated. It is recommended to evaluate other defense methods (e.g., composite and dynamic defenses) to support the conclusions.***
>
> We have expanded the experimental scope to include Dynamic Defenses and Composite Defenses. Using Gemini-2.5-Flash as the target model, we systematically evaluated In-Context Defense (ICD), Self-Reminder, and various composite defense strategies. We conducted a comprehensive horizontal comparison between CC-BOS and state-of-the-art attack methods (ICRT, GPTFUZZER). Detailed experimental results are shown in the following Table.
>
> The experiments demonstrate that CC-BOS exhibits strong adaptability when facing prompt-based dynamic defenses. Notably, under the highly effective Self-Reminder mechanism, GPTFUZZER's attack success rate drops directly to 0% and ICRT drops to 8%, while CC-BOS maintains an ASR of 28%. Under the ICD (one-shot) and (two-shot) defense measures, CC-BOS maintains ASRs of 76% and 54%, respectively.
>
> Furthermore, we constructed multiple composite defense measures. CC-BOS maintains an ASR above 16% across various composite defense settings. In the "ICD (2-shot) + Self-Reminder + LG (Output)" setting, ICRT and GPTFUZZER are rendered almost completely ineffective (ASR of 0% and 2%, respectively), whereas CC-BOS still maintains an attack success rate of 16%. These extended experiments fully validate the significant advantages of CC-BOS in coping with dynamic and composite defenses. For more details, please refer to Appendix [H].
>
> | Defense Strategy | CC-BOS (Ours) | ICRT | GPTFUZZER |
> |------------------|---------------|------|-----------|
> | No Defense | 100.00% | 92.00% | 28.00% |
> | **Dynamic Defenses** ||||
> | ICD (1-shot) | 76.00% | 46.00% | 24.00% |
> | ICD (2-shot) | 54.00% | 20.00% | 22.00% |
> | Self-Reminder | 28.00% | 8.00% | 0.00% |
> | **Composite Defenses** ||||
> | ICD (2-shot) + Self-Reminder | 24.00% | 6.00% | 6.00% |
> | ICD (1-shot) + Llama Guard (Output) | 22.00% | 0.00% | 20.00% |
> | ICD (2-shot) + Llama Guard (Output) | 16.00% | 0.00% | 16.00% |
> | Self-Reminder + Llama Guard (Output) | 18.00% | 0.00% | 0.00% |
> | ICD (2-shot) + Self-Reminder + LG (Output) | 16.00% | 0.00% | 2.00% |
>
> ---
>
> ***Q1: The paper claims that 'the security vulnerabilities of Classical Chinese stem from the unique challenges posed by its linguistic characteristics to alignment mechanisms.' This is confusing. Since alignment is difficult, why is the model still able to translate?***
>
> Thank you for this question. We acknowledge that our original phrasing may have been ambiguous. To address your concern, we provide the following explanation based on the mismatch between pre-training and alignment:
>
> **Pre-training Capability:**
> LLMs acquire robust comprehension and translation capabilities for Classical Chinese during the pre-training phase, as they are exposed to vast amounts of historical literature and corpora. This ensures the model intrinsically understands the syntax and semantics of the input.
>
> **Alignment Gap:**
> However, current safety alignment processes are predominantly concentrated on modern language. Consequently, while the model can read and understand the adversarial instructions, the safety defense layer fails to recognize the malicious patterns within this specific linguistic distribution.
>
> Thank you for this feedback. We have revised all the ambiguous statements in the Introduction accordingly.
>
> ---
>
> ***Q2: The calculation of the attack cost (Avg.Q) in Table 3 is confusing. The table reports an average of approximately 1.5 queries, whereas a single round appears to require at least 4 interactions. Please clarify and update.***
>
> Thank you for this keen observation regarding the calculation of Avg.Q. We sincerely apologize for the confusion caused by the lack of a precise definition in our initial submission.
>
> We would like to clarify that our reported Avg.Q follows the established conventions of prior works (e.g., PAIR, TAP, CL-GSO), which strictly count the queries sent to the Target LLM. We acknowledge the reviewer's correct assessment that the "generation, translation, and evaluation" steps involve additional queries; however, these were excluded from the Avg.Q metric to ensure fair comparison with baseline methods.
>
> We realize that failing to explicitly distinguish between "Target Queries" and total queries was a significant oversight. Following your suggestion, we have rewritten the relevant content in Section 4.1 (Experimental Settings) of the revision to clearly define the scope of Avg.Q. We are grateful for your feedback, which has helped us improve the rigorousness of our experimental presentation.

---

### Official Review · Reviewer_yuc5 · 2025-11-01

**Soundness:** 3
**Presentation:** 3
**Contribution:** 3
**Rating:** 6
**Confidence:** 3

**Summary:**

This paper introduces a novel black-box jailbreak attack on LLMs using the linguistic properties of classical Chinese. The proposed framework, CC-BOS, formalizes this attack into an 8-dimensional strategy space that combines jailbreak tactics with classical concepts and uses a bio-inspired algorithm to find optimal prompts. The method achieves a near-100% ASR against six LLMs on three safety benchmarks, demonstrating high efficiency and robustness.

**Strengths:**

1. The paper contributes a structured framework (CC-BOS) by formalizing the attack into a well-defined 8-dimensional strategy space.

2. The empirical evaluation is extensive, with near-perfect success across six SOTA models and three distinct benchmarks. The high query efficiency and robustness against Llama-Guard-3 underscore the potential of the attack.

**Weaknesses:**

1. The success of the proposed method is critically dependent on an “attack LLM” (Deepseek-Chat) to generate the final prompts. It’s unclear if the success comes from the 8D-space or simply this model’s specific generative ability.

2. The paper highlights Fruit Fly Optimization but fails to justify its use over other standard black-box optimizers (e.g., genetic algorithms, random search), rendering the optimizer’s specific contribution is unclear.

3. The defense experiment (Table 4) is limited to Llama-Guard-3-8B. A more compelling test would be using the proposed translation module as a defensive pre-processing step.

**Questions:**

Please address all concerns in the Weaknesses section.

---

> ### Author Response · Authors · 2025-11-20
> **Rebuttal by Authors [1/2]**
>
> Thank you for your valuable review and suggestions. Below we respond to the comments in Weaknesses (W) and Questions (Q).
>
> ***W1: The success of the proposed method is critically dependent on an "attack LLM" (Deepseek-Chat) to generate the final prompts. It's unclear if the success comes from the 8D-space or simply this model's specific generative ability.***
>
> We sincerely appreciate the reviewer for raising this critical question. The exceptionally high ASR of our method is primarily attributed to the proposed 8D Strategy Space, rather than the inherent capabilities of the specific attack model (Deepseek-Chat). To substantstantiate this claim, we provide evidence from both the ablation study in our main paper and additional generalization experiments:
>
> **1. Verification of Strategy Effectiveness (Based on Existing Ablation Study):**
> As demonstrated in Table 6 (Section 4.4) of our submission, we conducted a ablation analysis of the strategy components.
> - **Without Strategy:** When the attack model (Deepseek-Chat) was used to generate Classical Chinese prompts directly (Base) without our strategy guidance, the ASR on Claude-3.7 was only 18%.
> - **With Strategy:** When guided by our proposed 8D strategy space, the ASR significantly increased to 60% (and further reached 100% when combined with bio-inspired optimization).
>
> This significant gap confirms that the 8D strategy space is the decisive factor in unlocking attack potential, rather than the attack LLM's inherent capabilities.
>
> **2. Verification of Attack Model Generalization (New Experiments):**
> To further dispel concerns regarding the dependence on "Deepseek-Chat," we conducted additional experiments by replacing the attack model with GPT-3.5-Turbo and Gemini-2.0-Flash. We evaluated these against multiple target models, including Gemini-2.5-Flash, GPT-4o, and Deepseek-Reasoner.
>
> The results are shown in the following Table. It shows that our method consistently maintains an ASR of over 94%, even when the attack model is switched. This confirms that the high success rate stems from the 8D Search Space rather than the reliance on a single specific attack model. For further details, please refer to Appendix [E].
>
> | Target Models | Gemini-2.5-Flash | GPT-4o | Deepseek-Reasoner |
> |---------------|------------------|--------|-------------------|
> | **Attack Models** | **ASR** | **ASR** | **ASR** |
> | Deepseek-Chat | 100% | 100% | 100% |
> | GPT-3.5-Turbo | 98% | 100% | 96% |
> | Gemini-2.0-Flash | 94% | 96% | 96% |
>
> ---
>
> ***W2: The paper highlights Fruit Fly Optimization but fails to justify its use over other standard black-box optimizers (e.g., genetic algorithms, random search), rendering the optimizer's specific contribution is unclear.***
>
> Thank you for raising this important question regarding the choice of the optimization algorithm. We selected the Fruit Fly Optimization Algorithm (FOA) specifically for its significant advantage in handling high-dimensional discrete search spaces, enabling highly efficient convergence to optimal solutions.
>
> To fully justify this choice, we provide evidence from two perspectives:
>
> **1. Validation of FOA's Contribution (Existing Ablation):**
> As shown in Table 6 (Section 4.4) of our paper, utilizing the 8D strategy space alone yields an ASR of 60%. However, introducing FOA for search significantly boosts the ASR to 100%. This substantial improvement demonstrates that FOA effectively searches the vast combinatorial strategy space to lock in the most potent attack vectors.
>
> **2. Superiority over Standard Optimizers (New Comparative Experiment):**
> To demonstrate the superiority of FOA, we conducted a controlled experiment on GPT-4o, comparing FOA against Genetic Algorithm (GA) and Random Search. As presented in the following Table, FOA achieves a perfect 100% ASR, outperforming GA (94%) and Random Search (90%). More importantly, regarding efficiency, FOA requires only 1.28 queries on average to succeed. In contrast, GA requires 4.04 queries, and Random Search requires 6.10 queries. These results confirm that FOA is significantly more query-efficient than standard baselines, making it the optimal choice for black-box settings. Further details are provided in Appendix [F].
>
> | Optimizer | ASR (%) | Avg. Q |
> |-----------|---------|---------|
> | **FOA (Ours)** | **100%** | **1.28** |
> | Genetic Algorithm (GA) | 94.00% | 4.04 |
> | Random Search | 90.00% | 6.10 |

---

> ### Author Response · Authors · 2025-11-20
> **Rebuttal by Authors [2/2]**
>
> ***W3: The defense experiment is limited to Llama-Guard-3-8B. A more compelling test would be using the proposed translation module as a defensive pre-processing step.***
>
> We sincerely appreciate this valuable suggestion. Although our translation module was originally designed to enhance readability and evaluation accuracy, as you noted, it is indeed well-suited to serve as a pre-processing step for Output Defense.
>
> To verify this, we conducted additional comparative experiments. We integrated the translation module into the Llama-Guard output detection pipeline (specifically, translating the model's Classical Chinese responses into Modern English before safety inspection) and compared it against the standard defense.
>
> **Defense Efficacy via Translation:**
> The experimental data confirms that mitigating the linguistic obscurity of Classical Chinese effectively enhances defense capabilities. On Deepseek-Reasoner and Gemini-2.5-Flash, employing translation as an output pre-processing step resulted in a decrease in ASR compared to standard defenses. For further details, please refer to Appendix [G].
>
> | Defense Mechanism | Deepseek-Reasoner | Gemini-2.5-Flash |
> |-------------------|------------------|------------------|
> | Standard Output Defense (Raw Output) | 36.00% | 24.00% |
> | Standard Dual Defense (Input + Raw Output) | 28.00% | 22.00% |
> | Trans-Enhanced Output Defense (Translated Output) | 26.00% | 22.00% |
> | Mixed Dual Defense (Input + Translated Output) | 20.00% | 20.00% |

---

> ### Author Response · Authors · 2025-11-27
> **Looking forward to further feedback**
>
> Dear Reviewer yuc5,
>
> Sorry for bothering you, but the discussion period is coming to an end in a week. Could you please let us know if our responses have alleviated your concerns? If there are any further comments, we will do our best to respond.
>
> Best,
>
> The Authors

---

### Author Response · Authors · 2025-11-20
**Summary of Paper Revision**

We thank all reviewers for their constructive feedback, and we have responded to each reviewer individually. We have also uploaded a **Paper Revision** including additional results and illustrations:

* **Introduction** (Pages 2): Clarified the explanation of Classical Chinese vulnerabilities by highlighting that they result from a safety blind spot rather than limited data coverage.
* **Experiments** (Page 7): revised the definition of Avg.Q.
* **Appendix E** (Page 26): added evaluation on different attack LLMs.
* **Appendix F** (Page 27): added comparative analysis of optimization algorithms.
* **Appendix G** (Page 27): added evaluation of translation as a defensive pre-processing step.
* **Appendix H** (Page 27): added evaluation against dynamic and composite defenses.
* **Appendix I** (Page 28): added universality analysis across classical languages.

---

### Author Response · Authors · 2025-12-01
**Summary for AC: ICLR 2026 Submission 22564 - CC-BOS**

Dear PCs, SACs, ACs, and Reviewers,

Thank you very much for your valuable contributions to our work. To assist the newly assigned AC and help reduce their workload, we provide below a summary of the key points from the reviews and the reviewer-author discussions.

### Strengths
We are grateful that the reviewers recognized the novelty of our linguistic perspective and the effectiveness of our method. In particular:

* **Discovery of a novel linguistic vulnerability:** The revelation of the "High Capability - Low Alignment" mismatch in Classical Chinese as a critical safety blind spot. (kUsg, Qb4q)
* **Systematic Strategy Space:** The formalization of prompt generation into an 8-dimensional strategy space that effectively fuses jailbreak tactics with the character of Classical Chinese. (yuc5, Qb4q)
* **Efficient Bio-Inspired Optimization:** The novel application of a fruit fly-inspired algorithm (BIO) to efficiently explore the high-dimensional strategy space. (Qb4q, hZfy)
* **Superior Performance:** The method demonstrates superior performance, achieving nearly 100% ASR across six mainstream LLMs. (All reviewers)
* **Robust evaluation methodology:** A two-stage translation module ensuring objective safety assessment by mitigating linguistic ambiguity. (Qb4q)

### Concerns and Our Addressing
In response to concerns regarding generalization, optimization, and defense, we have supplemented corresponding experiments and analyses in both the revised main text and appendix, and explained them point by point in the rebuttal.
The following provides a more detailed summary by theme:

**Concerns about Algorithm Justification and Novelty (yuc5: Weakness 2; Qb4q: Weakness 3; hZfy: Weakness 1)**
* **Our Addressing:** We conducted new comparative experiments (Appendix F) demonstrating that FOA achieves 100% ASR with only 1.28 queries, significantly outperforming GA and Random Search. We clarified that distinct from AutoDAN’s token-level optimization via Genetic Algorithm (GA), our method optimizes a high-level 8-dimensional strategy space constructed from Classical Chinese linguistic features, utilizing Bio-Inspired Optimization (BIO) for efficient exploration.

**Concerns about Dependence on Attack Model and Generalization (yuc5: Weakness 1; Qb4q: Weakness 1)**
* **Our Addressing:** We conducted new experiments replacing the attack model with GPT-3.5-Turbo and Gemini-2.0-Flash. Our method maintained above 94% ASR (Appendix E), proving that efficacy stems from the 8D strategy space, not model-specific capabilities. We also successfully extended the framework to Latin and Sanskrit (Appendix I), achieving above 94% ASR on three mainstream LLMs.

**Concerns about Experimental Rigor and Defense Evaluation (yuc5: Weakness 3; kUsg: Weakness 1, Question 2; hZfy: Weakness 2)**
* **Our Addressing:** We expanded the evaluation to include Dynamic Defenses and Composite Defenses. CC-BOS maintained robust performance. We implemented a Translation-based Defense (as suggested by Reviewer yuc5), confirming that translating Classical Chinese outputs to Modern English can serve as an effective defense preprocessing step. We also clarified that "Avg. Q" counts queries sent strictly to the Target LLM and updated the manuscript to prevent ambiguity. Regarding concerns about 'prior knowledge' in examples (hZfy), we clarified that domain-specific terms (e.g., saltpeter) are autonomously generated by the model via our strategy, without revealing specific strategies or manufacturing procedures, ensuring a fair comparison.

**Concerns about Interpretability and Theoretical Mechanisms (hZfy: Weakness 3; kUsg: Question 1; Qb4q: Weakness 2)**
* **Our Addressing:** We articulated the "Capability-Alignment Mismatch": LLMs possess high capability in classical languages from pre-training but lack safety alignment, creating a defense blind spot. We detailed the specific mapping of linguistic features (e.g., metaphor, polysemy, authority bias) to our 8 strategy dimensions (Appendix C.2).

Due to the substantial expansion of our experiments and appendix to address reviewer feedback, our rebuttal and the revised version were submitted relatively late. Consequently, some reviewers may not have had sufficient time to carefully read the full rebuttal and the newly added experimental results, and may therefore not have updated their scores or continued the discussion accordingly. Nevertheless, we have made every effort to address all concerns point-by-point. Notably, Reviewer kUsg has already validated these efforts, confirming that their issues are resolved.

Above, we have faithfully summarized all reviewer comments and our corresponding responses, hoping that this will assist the AC's work. We are deeply grateful to the reviewers, AC, SAC, and PC, for their dedicated effort and excellent work. Their insightful feedback has further strengthened our paper. The authors offer their sincere respect and appreciation to all involved!

Sincerely,
Authors

---

### Meta-Review · Area_Chair_p25R · 2025-12-28

**Summary:**

This paper investigates the jailbreak attack problem of Language Learning Models (LLMs) and proposes a novel black-box jailbreak attack method, CC-BOS. This method constructs an eight-dimensional strategy space using the linguistic features of Classical Chinese and employs a fruit fly-inspired bio-optimization algorithm for efficient space exploration. Experimental results demonstrate good performance of CC-BOS. The method is reasonable, and its effectiveness has been verified experimentally. The use of Classical Chinese to generate adversarial prompts is insightful. Reviewers raised several issues regarding the paper's evaluation and writing, such as insufficient attack LLMs and defense methods used in the experiments, lack of experiment on method generalization to other languages, some ambiguous claims, unclear descriptions of differences from the related method AutoDAN, and unconvincing explanation of the rationale/reason for using Classical Chinese instead of other languages ​​in CC-BOS. The authors conducted supplementary experiments, reported new results, and made some clarifications in the rebuttal. The rebuttal has resolved most of the main issues. The experimental results and clarifications in the rebuttal section need to be added in the next version of the paper.

**Reviewer Concerns:**

The rebuttal has resolved most of the main issues raised by the reviewers, but the concern regarding unconvincing explanation of the rationale/reason for using Classical Chinese instead of other languages ​​in CC-BOS has not been fully addressed. The authors are encouraged to provide more evidences to show the advantage of using Classical Chinese over othe languages. I also doubt whether all eight strategy dimensions are useful for CC-BOS, or whether only some of them are effective.

**Reviewer Scores:**

The score of Reviewer Qb4q would have been changed.

---

### Decision · Program_Chairs · 2026-01-26

Accept (Poster)